# Verbal Instruction for Pelvic Floor Muscle Contraction among Healthy Young Males

**DOI:** 10.3390/ijerph191912031

**Published:** 2022-09-23

**Authors:** Noa Ben Ami, Ron Feldman, Gali Dar

**Affiliations:** 1Department of Physiotherapy, Faculty of Health Sciences, Ariel University, Ariel 4077625, Israel; 2Department of Physical Therapy, Faculty of Social Welfare & Health Sciences, University of Haifa, Mount Carmel, Haifa 3498838, Israel; 3Ribstein Center for Research and Sports Medicine, Wingate Institute, Netanya 4290200, Israel

**Keywords:** incontinence, pelvic floor muscles, transabdominal ultrasound, verbal instruction

## Abstract

Teaching Pelvic Floor Muscle (PFM) contraction is a challenging task for clinicians and patients, as these muscles cannot be directly visualized. Thus, this study’s objective is to compare the effectiveness of six verbal instructions for contracting the PFM among young men, as observed with transabdominal ultrasound imaging. Thirty-five male physiotherapy students, mean age 25.9 ± 1.9 years, participated in the study. A 6 MHz 35-mm curved linear array ultrasound transducer (Mindray M5) was placed in the transverse plane, supra-pubically, and angled 15–30° from the vertical plane. During crook lying, participants received six verbal instructions for contracting the PFM, with bladder base displacement and endurance evaluated. Following the instructions, “squeeze your anus”, “shorten the penis”, and “elevate the scrotum”, over 91% of the participants performed a cranial (upward) bladder base displacement. During instruction six, “draw in”, which involves breathing, the PFM, and the transversus abdominis, only 25% performed cranial bladder base displacement (*p* < 0.001), and the endurance was the lowest (*p* < 0.001). Our findings suggest that several simple verbal instructions can be used for teaching PFM contraction to young males. Moreover, two instructions should be avoided: “draw in” and the general instruction “squeeze your PFM”, as they did not produce effective elevation of the bladder base.

## 1. Introduction

The pelvic floor muscles (PFM) are vital to male genitourinary health [1,2], comprising deep and superficial parts. The deep PFM, in conjunction with the urethral and anal sphincters, work in coordination to promote urinary and bowel continence [3]; the superficial PFM function is maintaining erectile rigidity and the expulsion of the contents of the urethra [4].

PFM training is intended to improve urinary control by increasing the strength, endurance, and coordination of the pelvic floor muscles and functional activation of the external urethral sphincter [2,3,5].

Pelvic floor muscle training (PFMT) may prove helpful in a variety of clinical conditions: stress urinary incontinence that follows prostate surgery [6,7,8], postvoid dribbling [1], erectile dysfunction, ejaculation issues including premature ejaculation [1,2,8]. Performing PFMT before prostate surgery decreased the duration and severity of urinary leakage and improved surgical outcomes [9]. However, the value of conservative management of post-prostatectomy incontinence after radical prostatectomy remains uncertain [10].

The 2019 NICE guidelines recommended that men with stress urinary incontinence caused by prostatectomy should receive supervised PFMT for at least three months before considering other options [8].

Teaching PFM contraction is a challenging task for clinicians and patients, as these muscles cannot be directly visualized, and therapists cannot demonstrate their contraction. Also, PFMs are difficult to assess. Digital rectal examination is often used to evaluate PFM among men with good reproducibility [11,12], but this invasive technique might cause discomfort, resulting in a lack of cooperation and poor outcomes [11]. Trans-perineal ultrasound (TPUS) is another method commonly used to assess PFM. This method enables the evaluation of specific anatomical points, such as the striated urethral sphincter, levator ani, and bulbocavernosus muscles [13,14,15], and it allows simultaneous investigation of multiple pelvic floor muscles by measuring urethral displacement. During the examination, the ultrasound transducer is placed on the perineum; thus, the participants are required to remove their undergarments [13].

One method that minimizes the discomfort related to the above techniques is to use a transabdominal ultrasound (TAUS) to assess PFM function by measuring bladder base displacement during rest and contraction. The bladder base can be easily seen via TAUS and is supported by the PFMs and their fascia [16]. The contraction of the PFM causes cranial (upward) bladder base displacement. The amount of movement is considered an indicator of PFM contractility [16]. However, the elevation of the bladder base is achieved by shortening the Levator Ani, and as such, TAUS does not evaluate all PFMs but rather the Levator Ani in particular [17]. The advantages of TAUS are that it is comfortable for the patient, does not cause pain or embarrassment, and is a non-invasive, reliable technique for assessing PFM [11,16]. A recent trial [18] which compared TAUS and TPUS approaches in men after radical prostatectomy (*n* = 95) and used similar instructions as Stafford et al. [17], found no significant difference (*p* > 0.05) between the assessment methods. They concluded that both the TAUS and TPUS approaches are reliable, easy to utilize, and measure function over time [18]. The disadvantage of TAUS is that it requires a full bladder (as well as the TPUS method) and, as mentioned, does not asses the whole components of the PFM [11,17,19].

Another issue regarding teaching PFM contraction is related to the instruction provided by the examiner. Variations in verbal instructions may influence the direction and extent of the contraction. Most studies on PFM instruction have been conducted on women, and evidence regarding the most effective verbal instruction for PFM contraction is still lacking in the literature [20,21,22]. To the best of our knowledge, only one study examined the effect of different verbal instructions on the activation of PFM among men [17]. It used TPUS and surface electromyography (EMG) with 15 healthy men.

The current study aimed to determine the best verbal instruction to achieve cranial bladder base displacement among young men, as evaluated with TAUS. We hypothesized that the most effective verbal instruction would be “squeeze the anus” as it was previously found to be the most effective verbal instruction among women [22].

## 2. Methods

### 2.1. Participants and Setting

The present study protocol was reviewed and approved by the institutional review board of *** University, *** (Reg. No. AU-NBA-20180315). Informed consent was submitted by all subjects when they were enrolled. The rights of the subjects were protected. This clinical trial was registered at NIH ClinicalTrials.gov, number NCT ***.

Inclusion criteria were male physiotherapy students willing to participate in the study. The exclusion criterion was receiving physiotherapy treatment for pelvic floor disorder.

The participants were asked to complete self-reported questionnaires regarding demographic characteristics (age, weight, physical activity) and a Short Form International Consultation on Incontinence Questionnaire (ICIQ-SF) relating to symptoms of urinary incontinence [23].

Thirty-five male undergraduate physiotherapy students, mean age 25.9 ± 1.9 years, participated in the study. All students had previous knowledge of PFM anatomy as they studied it in the anatomy course.

The bladder base movement was evaluated using diagnostic ultrasound (Mindray M5) with a 6 MHz 35-mm curved linear array ultrasound transducer placed in the transverse plane, above the pubis, and angled 15–30° from the vertical plane. One investigator (GD) with experience in performing ultrasound imaging examined all the participants. The transabdominal ultrasound technique was chosen to reduce discomfort and increase participants’ compliance with the study.

### 2.2. Protocol

The ultrasound examination was conducted following a standardized bladder filling protocol, as per previous studies [16,22], thus, allowing clear imaging of the urinary bladder. This protocol required subjects to consume 600–750 mL of fluids over 1 h and complete the task 30 min before the ultrasound examination. The participants were asked to remove their shirts to uncover their lower abdominal area. They were examined in the supine position with their knees bent and with a pillow placed under their heads. The participants were given 6 different verbal instructions, in a random order, on how to contract their PFMs.

Instructions

Squeeze your pelvic floor muscles (“general instruction”)Squeeze your anusShorten the penisElevate the scrotumStop the flow of urineTake a moderate breath in, let the breath out, lift your pelvic floor muscles, and draw in your umbilicus. (“draw in”)

The same instructions commonly applied for PFM contraction in the professional literature were used [6,17,22]. The first instruction was general “squeeze your pelvic floor muscles”, as we wanted to explore if physiotherapist students could understand this general instruction. Instructions 2, 3, and 5 were taken from Stafford et al.’s trial [17], while instruction number 4, “elevate the scrotum”, was taken from Centemero et al. trial [6]. The sixth instruction is also very common and was investigated by Bø et al. [24]. This instruction includes breathing, PFM contraction combined with transversus abdominis (TrA) contraction. Although the order of contraction in our study was slightly different (PFM contraction followed by TrA contraction in our study compared with TrA contraction followed by PFM contraction in Bø et al. [24]), both instructions include expiration, PFM, and TrA muscle contraction.

A total of three contractions were performed for each verbal instruction. The first two were performed to evaluate the direction and displacement of the bladder base, while the third contraction was obtained to assess muscle endurance. For the first two contractions, the participants were asked to contract as strongly as they could and hold for three seconds with ten seconds of rest in between. The better of these two contractions were used for data analysis. For the third contraction, the participants were asked to contract and hold as long as they could, up to 20 s. The duration of contraction was measured in seconds.

PFM assessment, as evaluated by bladder base displacement, was performed via TAUS imaging. First, the examiner determined which direction of bladder base displacement was performed—implying cranial (upward) or caudal (downward) displacement of the bladder base. A marker with the on-screen cursor was placed on the image of the central portion of the bladder base at the junction of the hyper- and hypoechoic structures in the region of the greatest displacement visualized during a pelvic floor muscle contraction. Following this, the distance of displacement from resting to the contracted position was measured (in cm) using the on-screen calipers (Figure 1). The displacement was recorded, and the procedure was repeated, strictly maintaining the position of the US transducer for the entire testing procedure.

We measured the time that the bladder base displacement was maintained. The examiner marked the bottom of the bladder base using the on-screen cursor at the highest point achieved. The test ended when the bladder base started to descend from this point, and the time was recorded (in seconds). We referred to this as an endurance test.

The participants were instructed to contract PFM without contracting abdominal, gluteal, or adductor muscles and without performing posterior pelvic tilt. We also observed any changes in respiratory patterns as this may increase intra-abdominal pressure and lead to the caudal displacement of the bladder base [25].

The participants did not see the ultrasound images on the screen during the procedure. At the end of the examination, the patients were asked which instruction for performing a PFM contraction they perceived as the clearest. The study was not conducted in the English language.

For intra-test reliability, one of the authors (G.D.) re-measured the ultrasound images of ten individuals twice, two weeks apart. For inter-test reliability, the images of ten individuals were measured by an additional investigator blinded to the first author’s measurement results. We found intra- and inter-observer reliability to be high (intra-class correlations coefficient >0.8) for all measurement techniques used in this study. These values present substantial agreement, according to Landis and Koch (1977) [26].

### 2.3. Statistical Analysis

Statistical analysis was performed using SAS for windows version 9.4. Continuous variables were reported by means and standard deviations for normally distributed data or with median and interquartile ranges when data were not normally distributed. Categorical variables were reported according to relative frequencies.

A one-way, mixed-model repeated-measures analysis of variance (ANOVA) was used to examine the effect of verbal instruction on the three-outcome measures (displacement, endurance, and direction). The Logit link was used for the direction outcome, and the Gaussian link was used for the ranked transformed displacement and endurance outcome values. The studentized Maximum Modulus (SMM) post hoc adjustment method was used to reveal significance between pairs of conditions (different verbal instructions). A *p*-value of <0.05 was considered significant.

## 3. Results

The demographic characteristics of the participants are described in Table 1.

### 3.1. The Direction of Bladder Base Displacement

Analysis of variance revealed a significant difference in the number of participants who performed cranial bladder base displacement during the different instructions (*p* < 0.001). This was due to a significant difference (*p* < 0.001) between the direction of displacement during each instruction compared with instruction six, “draw in”, which resulted in the lowest percentage of participants performing cranial bladder base displacement.

About 94% of the participants performed cranial bladder base displacement during “squeeze your anus” and “shorten the penis” instructions. During the instruction of “draw in”, only 25% of the participants performed cranial bladder base displacement, and during the instruction “squeeze your pelvic floor muscles”, 69% of the participants performed cranial bladder base displacement (Table 2).

### 3.2. Bladder Base Displacement

The greatest displacement of the bladder base occurred during the “squeeze the anus”, “squeeze your PFM”, and “stop the flow of urine” instructions. The displacement was 0.9 cm (Table 2). The instruction “shorten the penis” and “draw in” resulted in the least displacement. The SMM post hoc analysis revealed that the greatest difference between pairs of instruction was between the displacements during “squeeze the anus” and “draw in” (*p* = 0.04).

### 3.3. Pelvic Floor Muscle Endurance

The mean duration of muscle contraction during most of the instructions was fourteen to fifteen seconds, except for the sixth instruction (“draw in”), for which the duration was significantly low at 8.3 s (*p* < 0.0001). Post hoc analysis showed a significant difference (*p* < 0.001) between muscle endurance time during each instruction compared with the sixth’ instruction, which exhibited the shortest muscle contraction.

### 3.4. Urinary Incontinence

Only one of the 35 participants reported urge incontinence of a small amount of urine.

### 3.5. Participants Preferred Verbal Instructions

Fourteen participants indicated that “stop the flow of urine” was their preferred instruction, and 10 participants chose “squeeze your anus”. All other instructions were less favored: three preferred “squeeze your PFM”, five preferred “shorten the penis”, one preferred “elevate the scrotum”, and two preferred “draw in”.

Agreement between preferred instruction and best bladder base displacement in the cranial direction occurred with only 12 (34%) participants.

## 4. Discussion

This study aimed to compare the influence of six verbal instructions for contracting PFM among young men, evaluated as displacement of the bladder base by TAUS. Most participants performed cranial bladder base displacement following the verbal instructions “squeeze your anus” (94.2%), “shorten the penis” (94.2%), “elevate the scrotum” (91.4%) and “stop the flow of urine” (88.5%).

Using the ‘general instruction’ (“squeeze your PFM”), only 69% of the participants performed cranial bladder base displacement, and using instruction six (“draw in”), only 25.7% were able to elevate the bladder base.

We believe that the relatively low number of participants performing correct activation during the number one instruction is a general instruction that is not clear enough for the participants. Thus, although being easy to use by the therapist, we suggest using more specific instructions.

The instructions “Shorten the penis” and “squeeze the anus” (#2+3) were best understood by most of the participants.

The “draw-in” Instruction was also evaluated by Bø et al. [24]. Bø et al. stated that instruction to contract PFM alone produces a significantly more effective PFM contraction than instruction, which involves breathing, PFM, and TrA contraction together.

It was previously found that during contraction of PFM, co-activation with the lower TrA muscle exists [24]. Consequently, an increase in IAP may occur. This causes a higher load on the bladder base, which presses it down and causes the puborectalis to lengthen [13]. This further reduces the contraction of the PFM. Stafford et al. [17] recommend that PFM exercises should be performed without increasing abdominal pressure. However, studies claim that abdominal muscle activity is a normal response to PFM exercise [27,28]; thus, co-activation of the muscles surrounding the abdominal cavity exists during PFM contraction. Also, advice to relax the abdominal wall when performing pelvic floor exercises is inappropriate and may adversely affect the performance of such exercises [29]. Following this debate, Thompson et al. [25] recommended that during the early stages of rehabilitation, abdominal contraction should not be performed as this may result in increased intra-abdominal pressure and caudal bladder displacement. However, during advanced stages of PFM rehabilitation, co-activation of TrA and PFM exercises should be included.

Following this debate in the literature and the present study, we should be careful when administrating the instruction of “draw in” as performing it inadequate will induce caudal bladder base displacement, which is not the desirable bladder base movement for PFM contraction.

Following all of the above, we can suggest that instruction six caused the shorter contraction time due to the possible elevation in intra-abdominal pressure, which increases the load on the PFM. Another explanation is that this instruction is very long and complex to understand as the participant needs to think about several components simultaneously (breathing, PFM, and Transversus abdominal muscle contraction) and, as such, complicated to conduct.

Regarding the amount of bladder base displacement via the TAUS, we found that bladder base displacement was greater in our study (9.1 mm) than in that of Kelly et al. [30] (5.4 mm). Endurance was slightly higher as well, at ~14 s compared with 11.9 s, respectively. Kelly et al. used the general instruction: ‘Draw in through your pelvic floor muscles as best as you can while breathing normally.’ This could account for the differences.

Most participants stated that the preferred verbal instructions were “stop the flow of urine” (fourteen participants) and “squeeze your anus” (10 participants). This result might be biased due to the embarrassment or discomfort of the students talking about intimate anatomy (e.g., the penis and scrotum).

Based on the results of the current study, we can suggest that several simple verbal instructions can be used for teaching PFM contraction to young men. This does not include the instruction of “draw in” as it did not produce effective elevation of the bladder base.

The current study gives some normative values for pelvic floor contraction in healthy, young males and offers a good starting point for more research that will compare these results to the patient population with urologic symptoms.

Men who need PFM strengthening can be taught to contract the PFM with several simple verbal cues without the need to remove clothing. The clinician should avoid using general instructions (e.g., “squeeze your PFM”) but rather specific instructions (e.g., “squeeze your anus”, “shorten the penis”, “elevate the scrotum”, and “stop the flow of urine”). In addition, the “draw in” instruction is also not a good instruction for teaching PFM contraction.

We know that there is a co-contraction between PFM and TrA muscle in every PFM contraction, but we believe that we should avoid instructing our patients to contract PFM with TrA and breathing verbally because it confuses them and should be done automatically.

### Limitations of the Study

The present study evaluated young male physiotherapy students, of whom 92% performed regular physical exercise. As such, our results may not be generalizable to a population of older men with urinary incontinence. However, it can give some normative values for pelvic floor contraction in healthy, educated, thin, young males.

As mentioned before, despite its advantages, TAUS’s main limitation is the inability to assess all parts of the PFM complex.

## 5. Conclusions

Our findings suggest that several simple verbal instructions (“squeeze your anus”, “shorten the penis”, “elevate the scrotum”, and “stop the flow of urine”) can be used for teaching PFM contraction to young males.

This does not include the instruction of “draw in”, which involves breathing, PFM, and transversus abdominis, as it did not produce effective elevation of the bladder base. We recommend avoiding using this or general instructions to contract the PFM and suggest using more specific instructions.

## Figures and Tables

**Figure 1 ijerph-19-12031-f001:**
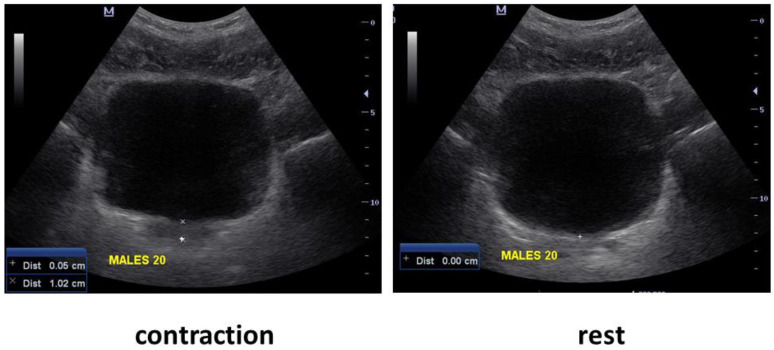
Ultrasound assessment of pelvic floor displacement at rest (**right**) and during contraction (**left**).

**Table 1 ijerph-19-12031-t001:** Baseline demographic and clinical characteristics of participants.

Variable *	Participants N = 35
Age (years)	25.97 ± 1.9
Married, n (%)	12 (34.2%)
Body mass index (BMI) †	24.6 ± 3
Smoke, n (%)	1 (2.9%)
Participate in regular physical activity, n (%)	32 (91.4%)
Perform pelvic floor muscle training regularly, n (%)	1 (2.9%)
Reporting urinary leakage about once a week or less often, n (%)	1 (2.9%)

* Values shown are mean (SD) unless otherwise noted. † Calculated as weight in kilograms divided by the square of the height in meters.

**Table 2 ijerph-19-12031-t002:** Differences in mean cranial displacement (cm) of the bladder base and muscle endurance between 6 instructions.

Variable	Instruction 1: Squeeze the Pelvic Floor	Instruction 2: Squeeze Your Anus	Instruction 3: Shorten the Penis	Instruction 4: Elevate the Scrotum	Instruction 5: Stop the Flow of Urine	Instruction 6: Draw in *	*p*-Value ** (Effect Size)
Number of participants with cranial bladder base displacement N (%)	24 (69%)	33 (94.3%)	33 (94.3%)	32 (91.4%)	31 (88.6%)	9 (25.7%)	<0.0001 (0.20)
Cranial bladder base displacement (cm), mean ± SD	0.91 ± 0.3 (N = 24)	0.91 ± 0.4 (N = 33)	0.79 ± 0.3 (N = 33)	0.87 ± 0.3 (N = 32)	0.90 ± 0.3 (N = 31)	0.82 ± 0.4 (N = 9)	0.015 (0.06)
Endurance (seconds) mean ± SD (N = number of participants who succeeded in holding the contraction	14.43 ± 6.0 (N = 23)	14.48 ± 6.7 (N = 31)	14.24 ± 6.7 (N = 33)	14.91 ± 6.1 (N = 32)	15.57 ± 6.7 (N = 30)	8.33 ± 8.4 (N = 7)	<0.0001 (0.14)

*** Draw in = instruction 6. **Take a moderate breath in, let the breath out, lift your pelvic floor muscles and draw in your umbilicus. **** *p*-values for ANOVA.**

## Data Availability

Not applicable.

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
