# Peer review of "Verbal Instruction for Pelvic Floor Muscle Contraction among Healthy Young Males"

_ijerph, 2022, doi:10.3390/ijerph191912031_

Round 1

Reviewer 1 Report

This study is clinically important because it forms the basis for studying correct and functional control of the pelvic floor muscles in men. It is easy to perform and understand by the patient and does not cause discomfort during an invasive examination. Well designed and executed and the authors presented the appropriate literature, and the conclusion in a careful but clinically clear manner. 

My recommendation is to “Accept” because I think this study is novel. Teaching pelvic floor muscle
(PFM) contraction for men (and women) is a challenging task for clinicians and patients, as these
muscles cannot be directly visualized, and therapists cannot demonstrate their contraction unless
they use digital rectal examination – and this invasive technique cause discomfort, resulting in a lack
of cooperation. This study novelty is to look for the best effective verbal instructions for contracting
the pelvic floor muscle among young men, using ultrasound so men who need PFM strengthening
can be taught to contract the PFM with several simple verbal cues, without the need to remove
clothing and pass a digital rectal examination.
Specific comments regarding the manuscript:
1. What is the main question addressed by the research?
The objective of the study was to compare the effectiveness of six different verbal instructions for
contracting PFMs among young men. They found four good instructions and two instructions that
should be avoided – this is important for all therapists to know – how to coach men to contract their
PFMs.
2. Do you consider the topic original or relevant in the field, and if so, why?
This topic is original. I didn’t see studies like this. And as a therapist, this knowledge is very important
and useful.
3. What specific improvements could the authors consider regarding the methodology?
I think the methodology is good.
4. Are the conclusions consistent with the evidence and arguments presented and do they address
the main question posed?
I think the conclusions are very straightforward and clear and can help clinicians.
5-6. References, tables, and figures are appropriate for this study.
I hope the report is more persuasive and will help the Academic Editor in reaching a final
decision

Reviewer 2 Report

Thank you for this well written paper on pelvic floor exercises in men.

Here are a few suggestions

Introduction:

Concern: the sec sentence does not follow from the first sentence; what is the problem in the first sentence.

Trans-peri- 46 neal ultrasound (TPUS) is another method commonly used to assess PFM. This method 47 enables the evaluation of specific anatomical points, such as the striated urethral sphinc- 48 ter, levator ani, and bulbocavernosus muscles [13–15] and it allows simultaneous investi- 49 gation of multiple pelvic floor muscles by measuring urethral displacement [13]. 50

One method that minimizes the discomfort related to the above techniques is to use 51 a transabdominal ultrasound (TAUS) to assess PFM function by measuring bladder base 52 displacement during rest and contraction.

Con 2: why does TAUS require a full bladder; is that different from the other ultrasound method

Clarify why you used TAUS

Results

For the determination of cranial base displacement – what was the distance of displacement and how did the distance vary.
